# Household Decision-Making Choices: Investment in Children’s Education or Self-Consumption

**DOI:** 10.3390/bs14030224

**Published:** 2024-03-10

**Authors:** Heng Jiang, Lili Liu, Yonglin Zhang

**Affiliations:** School of Statistics, Beijing Normal University, Beijing 100875, China

**Keywords:** human capital, utility model, continuous investment decision, maximum principle, transfer payment

## Abstract

Analyzing the sustainable decision-making mechanism between household consumption and education investment can theoretically develop education. This study uses the continuous-time utility model to demonstrate the independent characteristics of consumption and education investment, as well as the principle of decision incompatibility in the decision-making process of the utility maximization problem. Then, we establish a three-phase logarithmic utility model to obtain the intertemporal decision-making path of a family. The analysis shows that the investment allocation ratio between the two phases depends on the expected and discounted level of the offsprings’ abilities, while the total investment level is related to parental altruism. When parents, with foresight, factor in prospective transfer payments from progeny, the optimal decision is to maximize their children’s ultimate human capital within a given total investment. Education investment not only squeezes out consumption but also promotes consumption in various periods due to future transfer payments. The decision-making process of three typical growth stages indicates that as offspring mature and their human capital increases, parents’ willingness to invest in education decreases while self-consumption escalates. This study provides a new perspective and theoretical basis for studying household education expenditure, motivation, and related policy formulation.

## 1. Introduction

Education is an important part of human capital formation for economic growth, and investment in education has important future value [1,2]. Individuals acquire knowledge and skills through education, thereby enhancing the human capital required for engaging in economic activities [3]. The education choices of the family are one of the most important reasons for the heterogeneity of individual education paths [4]. From a micro-family perspective, parents, as the decision-makers in a typical family, can influence the amount and timing of their children’s education investments based on family wealth and their expectations [5,6]. Fiscal policies have significant influences on education investments by modulating household income levels [7,8]. They are the primary strategies adopted by governments to reduce educational inequality [9]. At the same time, there is no reverse causality between government education expenditures and household educational investments [10]. Adem [11] discusses that direct financial transfers to underprivileged families do not increase their will to acquire educational resources. Instead, demand-side cost-sharing mechanisms appear to incentivize parents more effectively. The difference in policy effects is due to household consumption preferences and demands [11]. Thus, a proper understanding of the behavioral motivation, expected goals, and achievement mechanisms of family investment in children’s education is of great value for formulating education policies from the perspective of individual families.

Self-consumption and investment in education are two parts of family expenditure [12]. Until children reach adulthood, their family is the main source of support for their growth. From the parents’ perspective, investing in their children’s education also requires meeting self-consumption needs. In reality, parents are not completely selfless. As a result, they do not pour all of their disposable income into raising their children. They always make trade-offs between consumption and investment. Education policymakers should not only consider how to promote parents’ willingness to invest in education. They should also pay attention to influencing the structure of consumption and investment at the micro-household level to make their policies more efficient [13].

The objectives of this study are to analyze in depth the process of family decision-making in which parents choose to allocate self-consumption and investment in their children’s education. It also analyzes the changes and causes of dynamic decision-making at different stages of their children’s lives.

Our findings may contribute to analyzing the household choice of investment in the accumulation of children’s human capital from a novel perspective. We consider the dual attitudes of parents’ egoism with consumption demand and altruism with care for children’s education. Our work breaks away from the constraints of traditional research that only focuses on maximizing human capital or policies to increase investment in education, allowing for a more vivid interpretation of family decision-making. It may explain why direct subsidies or policy incentives do not increase investment in family education equally. We analyze the strategies employed by parents with consumption needs in allocating educational investments across various stages of their children’s development.

The rest of the study is structured as follows: The first part qualitatively analyzes the circular decision-making between consumption and education investment and interprets the movement path and principles of family dynamic decision-making. The second part constructs a utility model to analyze parents’ decision-making choices, decision-making paths, and decision-making motivations. The third part displays household decision-making processes in the three typical stages of children’s growth, including preschool, compulsory education, and higher education. It interprets the reasons for the differences in household consumption choices and education investment decisions in different stages. Finally, the results of this study may also help policymakers use consumer and education policies together to promote household education investment.

## 2. Characteristics of Consumption and Investment Decisions in Households

In the study of economics, investing in children’s education is a form of consumption that generates future returns [14,15]. Parents are often faced with allocation decisions between self-consumption and investment in their children’s education. Consumption and investment constrain and influence each other, intending to obtain future income returns to provide higher consumption capacity [16]. Zhu and Yu verified that education expenditure significantly attenuates the SWB of Chinese households. This is due to a decline in consumption [17]. Education will have a crowding-out effect on self-consumption [17,18]. Therefore, parents in families will have a balance between consumption and education investment, and their investments in education are influenced by their willingness to consume.

The growth of children’s abilities through education brings a strong sense of happiness and fulfillment to parents. Under the influence of traditional culture, the younger generation rewards their parents with a transfer payment of future income. Self-satisfaction is an emotional utility. The progress of children makes parents feel proud and honored, thus generating happiness. This increases parents’ favoritism and altruism, and transfer payments directly increase parents’ lifetime budget constraints. Therefore, altruistic and exchange motives reflect that children’s growth has a dual value of consumption and investment for parents [19]. Although children grow faster because of the increase in education expenditures, the limited income of most families cannot fully meet the needs of their children’s growth. Especially in underdeveloped areas with low-income families, the income is more used for basic consumption in life. Thus, as parents choose their children’s education, it must be accompanied by a trade-off in self-satisfaction. This phenomenon is also used to explain the lower intergenerational mobility and higher persistence of education levels. High-income families can invest more in education [20,21,22,23], while low-income families have to meet their survival needs.

An increase in investment also reduces the incentive to consume. In a complete financial market, borrowing funds is no longer difficult, and individuals can make optimal consumption and investment choices based on their lifetime budgets rather than their current income [24]. Under this assumption, individuals can freely make their decisions and achieve higher utility, consumption, and other goals.

Becker et al. [5,22] discussed that some detailed discussion of family investment in education was made, highlighting the importance of the theory of family behavior for intergenerational inequality in education. He analyzed the important impact of intergenerational relationships on the process of human capital accumulation. Parents’ investment is influenced by budget constraints, altruistic factors, children’s endowments, consumption, and so on. The level of human capital will experience intergenerational convergence, and income inequality will continue to exist.

There are usually two different research perspectives on the impact of children’s growth on consumption and investment decisions in families. One emphasizes skill formation based on the process of human capital accumulation [25,26,27,28]. The optimal growth of children becomes possible through a rational allocation of investment. The other focuses on the income constraints and movement paths of economic consumption and investment, which is our perspective in the following discussion. For families, the level of income that can be invested in education is subject to a limited budget constraint. Usually, the household decides to maximize the utility of expenditures. At this point, the examination focuses on the optimal income distribution that maximizes individual utility rather than a separate human capital accumulation process.

Figure 1 maps the decision-making cycle in the family. Parents determine the level of investment and consumption in the current period based on their social environment, experience, and knowledge and examine their children. Then, they estimate investment and consumption for later periods. After the growth and consumption phases, at a new time, parents will continue to make a new round of decisions based on the current situation and continue the cycle. In this case, the parents’ decisions are forward-looking, and they will consider future situations to survive.

The decision-making process in Figure 1 shows dynamic changes in which parents flexibly choose their decisions over a continuous period and make the best judgments currently available. Notice that the choice of consumption in this process can affect the absolute level of education investment. The ability level of children will not affect parents’ current consumption decisions. In fact, there is a continuous feedback mechanism in a continuous decision-making process. State variables will not affect consumption and investment levels at the decision time but only affect future decision results.

This dynamic decision-making process exhibits significant heterogeneity at different time points. Parents’ decision-making behavior is continuously dynamic along the timeline of their children’s growth, with parents making new decision choices continuously, as shown in Figure 2. Heckman [27,29] points out that early human capital investment promotes later human capital investment. However, from the parents’ point of view, this promotion needs to satisfy their own judgment of the actual situation. In recent years, Su and Nirmala [30] find that parents are increasingly paying attention to investing in their children’s early years. Boneva and Rauh [31] use a randomized trial to analyze the presence of parental beliefs that make early investment more valuable and increase the efficiency of their children’s human capital accumulation. These illustrate the importance of the period of educational investment.

If intergenerational transfers from children exist in the future, parents’ aggregate budget constraints change and also raise parents’ expectations of their children. This transfer can be negative, but usually does not occur. This effect is reflected in the child feedback link in Figure 1. Mu and Du [32] find that social security also affects the level of transfers, yet parental preferences are often culturally determined. Becker [33] describes the composition of the members and their behavioral characteristics in the typical patriarchal family, constructing a framework for analyzing family behavior under altruism. Raut and Tran [34] constructed a model of intergenerational transfers and demonstrated that children’s transfers are a win-win behavior and that investment in education is more future-valuable. Qin et al. [35], using the OLG model, found that the optimal investment mechanism of parents is related to their children’s future transfer payments, and that their children’s developmental opportunities depend on their parents’ capital capacity. Becker et al. [5] proposed that parents may gain benefits from their children by influencing their children’s preferences and choosing to invest more.

In studies on underdeveloped regions, Xiao et al. [36] discussed that human capital accumulation and self-utility enhancement are all important channels for household investment in education to attenuate poverty. Fang and Hunag [37] point out that the level of human capital can positively promote rural investment in education, but the widening of the urban-rural gap has a negative impact on rural education expenditure. Family capital investment in education does improve children’s ability to progress [38], but access to education has a significant income burden for rural families [39]. Gu and Yang [40] point out that both parents’ expectations of their children’s education and income level can increase investment in education. Chi and Qian [21] point out that low-income households will spend a higher proportion of their capital on investment in education than higher-income households, but due to the difference in the absolute level, it does not increase the mobility of income levels. Belief and increasing investment are positive factors in improving children’s human capital level. Consumption beliefs among low-income households significantly restrain education investment decisions.

Therefore, recognizing the decision-making process of parents’ consumption and education investment can help to better understand how families choose investment processes to influence their children’s development. There is a richness of parental influences on investment choices in education discussed in classical studies, but little realization exists that one cannot single-handedly increase investment in education to make things better. Parents’ self-motivation to consume can make their decisions a choice between consumption and investment. Strategies that take into account both their own consumption and their contribution to their children’s education, with a planned allocation of the quantity consumed versus the quantity invested, are the most important starting point for the research of our work.

## 3. Modeling and Analysis

Firstly, a continuous-time utility model is constructed to describe the decision-making process between parents’ self-consumption and investment in children’s education. By maximizing the utility function, it is shown that the optimal consumption growth rate and the optimal investment growth rate of parents have the characteristics of independence for a given level of total consumption or total investment. It is also shown that the two decisions of focusing on the sum of the utilities of children’s human capital outputs in each period and focusing on the final level of children’s human capital generally cannot coexist. The conclusions are then based on the use of a three-period utility model to calculate inter-period optimal decisions. Finally, the model incorporates children’s future transfers and draws conclusions about revealed preferences for decision-making under different situations.

### 3.1. Continuous-Time Utility Modeling

Assuming that a typical family consists of parents with stable jobs and one child, at the moment *t*, the level of consumption of the parents is $c_{t}$, the level of expenditure on investment in the education of the child is $k_{t}$, and the level of wages used for consumption and investment in education $w_{i}$. In the classical theory of human capital, when the study period is [0, *T*], the equation of human capital accumulation by children has the following form
(1)$${\dot{h}}_{t}=F(h_{t},k_{t})-\delta h_{t}$$

${\dot{h}}_{t}$ is the incremental human capital at time *t*. $F_{t}=F(h_{t},k_{t})$ means a human capital production function. *δ* is the level of human capital depreciation, and $k_{t}$ is the level of parental investment in education. Function *F* may have a more complex structure. It can deal with endogenous factors such as family time investment and school investment. These favorable factors for enhancing children’s growth can be simplified into exogenous technological progress and the increase of children’s environmental endowment.

When there is no borrowing, per-period parents have a strict budget constraint is:(2)$$\sum_{i=0}^{t}\left(c_{i}+k_{i}\right)\leq \sum_{i=0}^{t}w_{i}$$

In a complete financial market, parents are able to borrow money and make decisions freely. In reality, with the help of others, parents are usually able to make such behavioral choices. Here, we assume that the average family budget cannot exceed lifetime income. The total lifetime income that can be used for investing in education and self-consumption is *W*. This thus yields budget constraints.
(3)$$\int_{0}^{T}e^{-rt}\left(c_{t}+k_{t}\right)dt=\int_{0}^{T}e^{-rt}w_{t}dt\equiv W$$

*r* is the discount parameter. Parents maximize their utility function by choosing future consumption paths $\left\{c_{t}\right\}$ and investment paths $\left\{k_{t}\right\}$.
(4)$$\max U_{parent}=\underset{c_{t},k_{t}}{\max}\int_{0}^{T}\left(e^{-\rho t}U_{1}(c_{t})+\eta e^{-\rho t}U_{2}(RF(h_{t},k_{t}))\right)dt$$

The total utility of parents consists of the consumption utility function $U_{1}(c_{t})$ and the utility function of the new growth component of children’s human capital $U_{2}(RF(h_{t},k_{t}))$. It satisfies ${U_{1}}^{\prime }(\cdot )>0$, ${U_{1}}^{\prime \prime }(\cdot )<0$, ${U_{2}}^{\prime \prime }(\cdot )>0$, and ${U_{2}}^{\prime \prime }(\cdot )<0$. ρ denotes the subjective discount rate of utility. *R* denotes the value of a unit of human capital. η is an altruistic parameter that reflects the degree of utility preference between self-consumption and investment in education. It may also reflect the relative value of the utility of consumption and investment. $U_{2}$ also includes parents’ well-being.

The linear assumption of the value of human capital follows the classic theory of Ben Porath [41]. And our discussion focuses on the paths of consumption paths $\left\{c_{t}\right\}$ and investment paths $\left\{k_{t}\right\}$. In each period, human capital additions give parents the satisfaction and utility of investment. If parents derive greater well-being from their children’s education over a period of time, they decide to invest more in education. This change is given by Equation (4) $\eta e^{-\rho t}U_{2}(R{\dot{h}}_{t})$, which means η increases or $U_{2}$ has a higher marginal value for the level of children’s human capital. Therefore, the marginal utility of consumption and education investment is equal when maximizing Equation (4). It means parents choose a decline in consumption and an increase in education investment. Obviously, parental entertainment consumption is also part of total consumption. If they have higher entertainment utility, then η in Equation (4) decreases, which implies parents choose higher self-consumption with less investment in children’s education.

In anticipation of the future, the forgotten regression is usually ignored by the parents, and the human capital invested in the next period when the children are educated is the net of the forgotten regressions. Then, the regression is seen as a depreciation of human capital. Equations (1)–(4) constitute the main model.

The objective of Equation (4) reflects the fact that the utility of parents consists of two components: the utility of consumption and the utility of the new growth of children’s human capital in each period. Here, the growth of children can be interpreted as the return on investment in the current period. The *η* in Equation (4) can also be interpreted as the degree of parental motivation to invest in education, and the larger *η* is, the higher the utility that parents get from investing in their children’s education. The behavior that optimizes Equation (4) to maximize the total utility of the expenditure is called “Decision one”.

The total utility Equation (4) takes on a different form when the parents’ concern is the ultimate level of their children’s human capital. It means parents have no clear preference for the time of the formation of competence in their children’s lives. The only aim is to see the best level of their children’s human capital $h_{T}$ at moment *T*. The choice of the path is called “Decision two”. The utility function takes the form
(4a)$$\max U_{parent}=\underset{c_{t},k_{t}}{\max}\int_{0}^{T}e^{-\rho t}U_{1}(c_{t})dt+\eta e^{-\rho T}U_{2}(Rh_{T})$$

In addition, Equation (4) has a deformation as
(4b)$$\max U_{parent}=\underset{c_{t},k_{t}}{\max}\int_{0}^{T}\left(e^{-\rho t}U_{1}(c_{t})+\eta e^{-\rho t}U_{2}(R{\dot{h}}_{t})\right)dt$$

Equation (4) is used to describe the utility generated by the overall progress of children relative to the previous period. The interest is in the addition of children’s human capital, not the value-creating function of capital. Typically, depreciation is not observed, and parental decision-making usually considers the effects achieved by education inputs rather than a comprehensive evaluation. Thus, both functions have important uses. In particular, human capital can be expressed as an integral form of multi-period accumulation.
(5)$$h_{T}=h_{0}+\int_{0}^{T}{\dot{h}}_{t}dt=h_{0}{\left(1-\delta \right)}^{T}+\int_{0}^{T}{\left(1-\delta \right)}^{T-t}F(h_{t},k_{t})dt$$

Equations (4) and (4a) can capture the dual choice of consumption and investment in parental decision-making. The difference is that Equation (4) captures the instant gratification utility to parents from the increase in their children’s human capital. The decision-making behavior typically describes parents’ planned choice of investment in their children’s future education at various stages, and the value of the returns to achieving their investment goals at each period. Equation (4a) describes the process of education and development of their children, where parents do not care about the timing of their children’s acquisition of competences. These parents are more concerned about the expected level of their children’s human capital and will choose the best investment path for their children’s growth under a defined investment spending plan.

In contrast, the investment in education in Equation (4) is an investment process with continuous returns, whereas Equation (4a) resembles a phased forward investment with special returns in the future. From a parental perspective, investment and consumption are not completely free substitutes, and the decision-making process often pre-determines the total amount of investment or the total amount of consumption. The point of the decision is to achieve a higher level of self-consumption and a better upbringing for the children.

### 3.2. Principles of Decision-Making in Consumption and Investment

Under complete financial markets, using Equation (3) as a constraint, the Hamiltonian function of Equation (4) is
(6)$$\begin{array}{ll} \hat{H}(t,h_{t},k_{t},c_{t}) & =\;e^{-\rho t}U_{1}(c_{t})+\eta e^{-\rho t}U_{2}(RF(h_{t},k_{t}))\\ & +\;\lambda (t)(F(h_{t},k_{t})-\delta h_{t})-\mu (t)e^{-rt}(c_{t}+k_{t}) \end{array}$$

The maximization conditions are
(7)$$\frac{d\hat{H}}{d\lambda (t)}={\dot{h}}_{t};\;\;\frac{d\hat{H}}{dc_{t}}=0;\;\;\frac{d\hat{H}}{dk_{t}}=0;\;\;\frac{d\hat{H}}{dh_{t}}=-\dot{\lambda }(t);\;\;\dot{\mu }(t)=0$$

The transverse condition is $\lambda \left(T\right)=0$. Then, we calculate $d\hat{H}/dc_{t}$
(8)$${U_{1}}^{\prime }(c_{t})=\mu (t)e^{(\rho -r)t}$$

Notice that $d^{2}\hat{H}/{dc_{t}}^{2}=e^{-\rho t}{U_{1}}^{\prime \prime }(c_{t})<0$, thus, Equation (8) is a necessary condition for the maximization of Equation (4). $U_{1}$ includes consumption only, so ${\dot{c}}_{t}/c_{t}$ only relate to time *t*. We can draw an important conclusion.

**Conclusion** **1.** *Under the basic assumptions, when maximizing the objective Equations (4) and (4a), the growth rate of the parents’ optimal consumption path* $\left\{c_{t}\right\}$ *is relevant to time t only.* $\left\{c_{t}\right\}$ *isn’t dependent on the level of human capital of the offspring* $\left\{h_{t}\right\}$ *and the parent’s investment path* $\{k_{t}\}.$ *Once the optimal initial consumption level* $c_{0}$ *is chosen, the optimal consumption path* $\left\{c_{t}\right\}$ *is also determined*.

Then, Corollary 1 can be described as follows:

**Corollary** **1**  **(Principle of independent decision-making).**
*The optimal paths of consumption and investment in education can be decided independently at decision time t, when either the total consumption or the total investment is selected.*

The total level of consumption is the factor that affects the family education investment decision. Corollary 1 expresses the fact that the optimal investment in children’s education can be transformed into a local optimization problem after parents have chosen an initial level of consumption. The opportunity cost of consumption and investment is the marginal cost of the entire decision-making period. Because the level of consumption is set, the level of investment ability *K* is only related to the level of initial investment and lifetime income. We make
(9)$$\int_{0}^{T}e^{-rt}k_{t}dt=W-\int_{0}^{T}e^{-rt}c_{t}dt\equiv K(c_{0},W)$$

Assuming that the consumption path is given, then maximizing Equation (4a) at this point is also equivalent to maximizing the final level of children’s human capital. For the two alternative decisions of maximizing the sum of individual utilities in each period or maximizing the expected utility of the children’s human capital, the limit method is used to obtain Conclusion 2.

**Conclusion** **2**  **(Principle of incompatibility of decision-making).**
*Under the basic assumptions, the optimal investment paths obtained under the maximizing objective Equations (4) and (4a) will not be identical, given the level of total parental consumption.*

**Proof** **of** **Conclusion** **2.** A simple proof process is stated as follows: If the utility functions of the two decisions are the same, from the first-order condition
(10)$$e^{(r-\rho )t}\frac{d\int_{t}^{T}U_{2}(RF(h_{s},k_{s}))ds}{dk_{t}}=e^{(r-\rho )T}\frac{dU_{2}(RF(h_{T},k_{T}))}{dk_{T}}$$
(11)$$e^{rt}\frac{dU_{2}(Rh_{T})}{dk_{t}}=e^{rT}\frac{dU_{2}(Rh_{T})}{dk_{T}}$$We substitute two equations into Equation (5).
(12)$$\int_{t}^{T}(e^{\rho (T-s)}\frac{dU_{2}(RF_{s})/dF_{s}}{dU_{2}(Rh_{T})/d(h_{T})}-{\left(1-\delta \right)}^{T-s})\frac{dF_{t}}{dk_{t}}ds=0$$The above equation holds for any time *t*. Then, $U_{2}{\left(RF_{t}\right)}^{\prime }={\left(1-\delta \right)}^{T-t}e^{-\rho (T-t)}U_{2}{\left(RF_{T}\right)}^{\prime }$ or $dF_{t}/dk_{t}=0$. The former implies that, for a given growth path, the utility function is only a specific function of time. However, the parameter of the utility function in the generalization discussion is free, and the utility function cannot be just a function of time *t*. Moreover, the latter implies zero inputs due to the cumulative nature of human capital. The contradiction is obvious. □

Conclusion 2 illustrates that, in most cases, the future growth path of investing in education will vary depending on the decisions. Usually the two decisions cannot coexist, so the choice of decision can even show a display of preference in different discussions. In particular, it is shown that the elimination of the altruism parameter in the first-order conditional Equations (10) and (11) also suggests that altruistic choice means a trade-off between consumption and investment in education and that the structure of investment over time is irrelevant. Parents’ preference for their children’s education only determines the distribution of total consumption and total investment in education.

So far, we have constructed a continuous-time utility model under two decisions and obtained two important conclusions by mathematical means. And when the total investment or consumption is selected, it is clear from Corollary 1 that the utility level is at least locally optimal under different decisions.

### 3.3. Three-Period Logarithmic Utility Models and Decision-Making Mechanisms

In the decision-making process, intertemporal choice is an important element. Based on the assumption that the continuous-time model is difficult to solve, a three-period logarithmic utility model is developed to analyze the problem in order to reflect the inter-period comparison.

Assuming a classic intergenerational structure where a family chooses to invest in the education of a child, it will go through three stages of consumption and investment, as shown in Figure 3. Stage 1 and stage 2 parents will consume and invest in their child’s education, and the child will learn to increase his or her level of human capital but will not generate income, usually by attending school. In the third stage, children rely on their level of human capital to generate income, and parents only need to satisfy self-consumption. Usually, the initial level of human capital of the child is not zero.

Based on the structure of Equation (4), a logarithmic utility function is used. The objective function for maximizing utility is
(13)$$\underset{\Omega }{\max}H\equiv \underset{\Omega }{\max}\{\ln(c_{1})+\rho \ln(c_{2})+\rho^{2}\ln(c_{3})+\eta (\ln(RF_{1}(h_{1},k_{1}))+\rho \ln(RF_{2}(h_{2},k_{2})))\}$$

Here, $\Omega =\{c_{1},c_{2},c_{3},k_{1},k_{2}\}$, parents have three period consumption variables $c_{1}$, $c_{2}$, $c_{3}$ and two period investment variables $k_{1}$, $k_{2}$. *R* is the value of a unit of human capital. η is the altruism parameter. *ρ* is the discounted level of utility. And *r* is the discounted level of capital. *W* is the level of total income available to the parent for investment and consumption in the three periods, and the budget constraint is
(14)$$c_{1}+rc_{2}+r^{2}c_{3}+k_{1}+rk_{2}=W$$

In the first two stages of human capital formation, simply, the growth of children is an output of the investment in education, and the effects of the individual state in both stages are treated exogenously. We set that $F_{1}={A_{1}k_{1}}^{\alpha_{1}}$ and $F_{2}={A_{2}k_{2}}^{\alpha_{2}}$. So, the net increase in the level of human capital in the two periods is ${\dot{h}}_{1}={A_{1}k_{1}}^{\alpha_{1}}-\delta h_{0}=h_{1}-h_{0}$ and ${\dot{h}}_{2}={A_{2}k_{2}}^{\alpha_{2}}-\delta h_{1}=h_{2}-h_{1}$. Where $A_{1}$ and $A_{2}$ are composite parameters of capacity, *δ* is the depreciation rate, and $h_{0}$ means the initial human capital level of the offspring. At the decision point, where parents are not fully certain of their children’s future potential, the process of human capital growth is perceived as an act of pre-investment, and the outcome of children’s growth generates satisfaction and utility for parents. In particular, the production function *F* may have a richer form, and the calculation will be more complicated [42].

Constructing a Lagrange function
(15)$$\begin{array}{r} L=\ln(c_{1})+\rho \ln(c_{2})+\rho^{2}\ln(c_{3})+\eta (\ln(R{A_{1}k_{1}}^{\alpha_{1}})+\rho \ln(R{A_{2}k_{2}}^{\alpha_{2}}))\\ +\lambda (W-c_{1}-rc_{2}-r^{2}c_{3}-k_{1}-rk_{2}) \end{array}$$

λ is a Lagrange factor. Because $dL/dc_{i}=0$
(16)$$\frac{1}{c_{1}}=\frac{\rho }{c_{2}r}=\frac{\rho^{2}}{c_{3}r^{2}}=\lambda$$

Combining Conclusion 1 with Equation (16), in the parent maximization decision, the consumption growth rate is equal to (ρ−r)/r. Once the parents have determined the appropriate initial level of consumption, the future plan of consumption is also determined, independent of the investment path. Continuing with the calculation of the maximization condition
(17)$$\begin{array}{l} \frac{dL}{dk_{1}}=\alpha_{1}\eta \frac{1}{k_{1}}-\lambda =0\\ \frac{dL}{dk_{2}}=\alpha_{2}\eta \rho \frac{1}{k_{2}}-\lambda r=0\\ \;\frac{k_{1}}{k_{2}}=\frac{\alpha_{1}r}{\alpha_{2}\rho } \end{array}$$

Evidently, $d^{2}L/{dk_{1}}^{2}<0$ and $d^{2}L/{dk_{2}}^{2}<0$. Equation (17) is the maximization condition of L. The investment decisions at this point can explain a phenomenon. In the early years of a child’s life, if parents believe that their child’s human capital growth results from the accumulation of knowledge and skills that investment can provide, then the optimal investment decision depends on the degree of parental preference for the child η, elasticity of capital output *α*, and discount rates *ρ* and *r*. Because human capital factors such as the ability and potential of the offspring are not fully expressed at the initial stage, parents’ future investment decisions depend only on the characteristics of the current market environment and the initial level of consumption, and they aim to maximize the utility value of their investments. When the total amount of investment is given, parents choose their investment weights based on the expected capital output capacity *α* at different stages of the process, rather than the unknown level of their children’s capacity *A*.
(18)$$c_{1}=\frac{W}{1+\rho /r+\rho^{2}/r^{2}+\alpha_{2}\eta \rho /r+\alpha_{1}\eta }$$

At this point, the level of consumption, the level of investment, and the level of human capital in the objective function are uniquely determined. $dc_{1}/d\eta <0$ reflects the negative correlation between parents’ consumption levels and care for their children. $d^{2}c_{1}/d\eta^{2}>0$ reflects the decelerating decline in personal consumption when parents are more inclined to care for their children. $dk_{1}/d\eta >0$ and $d^{2}k_{1}/d\eta^{2}<0$ shows investment levels will show decelerated growth.

Parental concern increases the expectations of their children, but is still influenced by self-consumption, with a marginal diminution in the growth of investment. In particular, entertainment consumption can increase parental satisfaction. In different periods, parents’ preference for entertainment may lead to the decline of η. At this time, consumption $c_{i}$ rises and investment $k_{i}$ falls. Families make decisions that take into account market characteristics, the expected learning ability of their children, and parental preferences for their children, which include growth rates and initial level choices.

From Corollary 1, the optimal behavior of investment is considered locally under a stable consumption path. Based on Equation (4a), the objective function becomes:(19)$$\underset{\Omega }{\max}H\equiv \max\{\ln(c_{1})+\rho \ln(c_{2})+\rho^{2}\ln(c_{3})+\eta \rho \ln(Rh_{2})\}$$

Maximizing the objective function
(20)$$\ln k_{1}=\frac{1-\alpha_{1}}{1-\alpha_{2}}\ln(rk_{2})+\frac{1}{1-\alpha_{2}}\ln\frac{(1-\delta )A_{1}\alpha_{1}}{A_{2}\alpha_{2}}$$

When $\alpha_{1}=\alpha_{2}=\alpha$, it means: $k_{1}/k_{2}=r{\left((1-\delta )A_{1}/A_{2}\right)}^{\alpha -1}$.

A comparison of the two decisions reveals that the allocation ratio of investment is different. Because of the independent decision-making characteristics of consumption behavior and investment growth rate, global maximization of utility only requires choosing the appropriate initial level of consumption or investment. The optimization Equations (15) and (19) must satisfy that the marginal utility levels of consumption and investment are equal. Thus, the calculation can be obtained when choosing the global maximizing consumption and investment paths under the same conditions:(21)$$\frac{dH}{dc_{1}}=\frac{dH}{dk_{2}}<\frac{dL}{dk_{2}}=\frac{dL}{dc_{1}}$$

Equation (21) shows that the marginal utility of consumption in Decision 1 is greater than that of Decision 2, which then suggests that Decision 1 invests more in education and consumes less. One explanation is that Decision 1′s investment in education is inefficient for children. Parents lose the expectation of their children’s growth in order to satisfy the self-consumption utility. This illustrates that period-planned education decisions are not always optimal, and that sometimes temporary mediocrity is for the sake of better growth in the future. Here, there is uncertainty about the final level $h_{2}$ under both decisions due to the initial level of human capital.

From Equation (17), the proportion of investment allocated to the two periods of Decision 1 is independent of the child’s individual trait ability *A* and depends only on the ability *α* to use investment. Investment in this decision is treated as future multi-period consumption, and the output of human capital is the return on the productive utility of the investment in each period. In Decision 2, since the technical parameters $A_{1}$ and $A_{2}$ are uncertain, the structure of the function is influenced by the objective environment, current knowledge, perception of self, etc. Comparing the two decisions shows that when the level of investment under Equations (17) and (20) is different, parents in Decision 2 lose the intermediate period utility of their investment but allow their children to grow to maximize future expectations.

Take constraint Equation (20) into Equation (19) and derive $c_{1}$. It has two typical situations. When children are at high levels of human capital, there is a sufficiently large $h_{0}$ to make $dH/dc_{1}>0$. In the feasible domain of $c_{1}$, the optimal solution is $\max\{c_{1}\}$. It means children’s human capital is high, and parents will choose to make their own consumption without excessive investment in education. When $h_{0}$ is small, there is an optimal solution for the level of consumption. Parents will choose a suitable level of consumption and investment in education at the same time.

In short, in different growth stages, the marginal income of education decreases, and the optimal decision of parents will dynamically tend to choose self-consumption.

### 3.4. Future Transfer Payments

In the previous discussion, the budget constraint for parents was the level of total income available to the individual *W*. Variable *W* does not take into account the impact of child-related income. As kinship and care generate intergenerational income transfers, transfers from children are included in the model to analyze the impact on parental decisions.

In the future, children will spend part of their income on their parents’ retirement at $T_{3}$. Generally, the higher the level of children’s human capital is, the more they can earn. And more transfer payments will be made. Assuming that transfers from children are $b=\varphi h_{2}$, φ denotes the transfer payment of unit human capital. The parent’s budget constraint Equation (14) becomes
(22)$$c_{1}+rc_{2}+r^{2}c_{3}+k_{1}+rk_{2}=b+W$$

Obviously, the level of transfers should also be within the children’s earning capacity. φ/R is usually low in real terms because of the children’s own consumption, investment, and other needs. Assuming that both parents have positive expectations of transfer payments, transfers from children are positive incentives. It directly increases parents’ lifetime budget constraints. It increases both the level of consumption and investment in education in each period. From Equation (18), for each unit of transfer payments, total parental consumption increases $\frac{1+\rho /r+\rho^{2}/r^{2}}{1+\rho /r+\rho^{2}/r^{2}+\alpha_{2}\eta \rho /r+\alpha_{1}\eta }$ and total parental investment increases $\frac{\alpha_{2}\eta \rho /r+\alpha_{1}\eta }{1+\rho /r+\rho^{2}/r^{2}+\alpha_{2}\eta \rho /r+\alpha_{1}\eta }$.

An increase in the level of expected transfers is accompanied by parents’ preference for investing in their children’s education, and as care for their children increases, parents allocate a portion of their future income to less consumption and more investment. Thus, in anticipation of future transfers for their children, parents will maximize upfront investment and consumption. This decision-making behavior of using present consumption in exchange for future returns increases both their self-consumption and their children’s human capital.

From Equation (1)
(23)$$h_{2}={\left(1-\delta \right)}^{2}h_{0}+(1-\delta )F_{1}(k_{1},h_{1})+F_{2}(k_{2},h_{2})$$

Equation (23) is an implicit function of $h_{2}$ and $h_{0}$. At this point, we discuss a steady-state scenario with a given *φ*. Parents choose to rationally allocate the portion of future benefits given the future transfer ratios. $dh_{2}/dh_{0}>0$ means higher levels of initial human capital are associated with higher levels of future human capital, as well as higher returns to parents. Sometimes the initial level of human capital cannot be evaluated, and judgments based on the inheritance of traits, children’s performance, and life experience produce high evaluations that stimulate parents’ propensity to invest more.

When the objective decision is Equation (4), maximizing the utility of the parent will fail to maximize the level of human capital $h_{2}$, as described earlier. At this point, the proportion of investments used for both periods can be adjusted to meet Equation (20). The child will have a higher level of human capital with higher transfers in the future. Thus, the budget of the father’s generation will increase further, but there will be no decline in the level of self-consumption. Parents can increase investment until the equilibrium in Equation (22) is satisfied and additional self-consumption is available. It also means that using Decision 2, parents can obtain a greater level of self-consumption, or their children can reach a higher level of human capital. Especially, parents’ consumption profiles or children’s human capital are expected to become better.

**Conclusion** **3** **(Revealed preference).** *Assuming that the child is expected to give fixed transfers proportional to his own human capital, when parents have already decided on the level of total consumption or total investment, parents reveal their preferences for Decision 2.*

Conclusion 3 is presented in two scenarios. In the first scenario, when the optimal investment ratio of Decision 1 satisfies Equation (20), given the parents’ consumption level, the optimal decision maximizes the parents’ total consumption utility while the children are able to obtain a high level of human capital in the future, which is a win-win outcome. In the second scenario, when the conditions are not met, parents use Decision 2 to make adjustments. The consumption level and the children’s human capital are expected to be better when the situation does not get worse. In both cases, parents can obtain child transfers to improve budget constraints.

**Corollary** **2.** 
*After parents have decided on the consumption path, the optimal decision to maximize the expected level of children’s human capital is to make the two-period level of investment in children satisfy Equation (20) and make Equation (22) hold.*


**Corollary** **3.** 
*After parents have decided on the total amount of future investment, the optimal decision must be such that the two-period investment satisfies Equation (20) and parents can obtain the maximum utility level of total consumption.*


Two corollaries suggest that the advent of transfers makes it possible for parents to be willing upfront, not only to increase their investment in their children, but also to increase their own consumption. Conclusion 3 reflects the important value of focusing on the education of children. When there are intergenerational exchanges proportional to children’s incomes, it is preferable for parents to invest wisely in their children in order to maximize $h_{2}$ rather than satisfy self-consumption utility. Conclusion 3 can be generalized to some extent. When parents wish to obtain greater self-consumption utility from a fixed investment in education, the local decision path of investment must be maximizing the human capital of children. The future transfers are favorable to parents. Otherwise, parents can benefit from adjusting their decision by adjusting the level of investment so that their children will give more transfers in the future.

When the level of transfers *b* is constant or independent of the level of children’s human capital, there are no deterministic revealed preferences for the two decisions based on maximizing Equations (4) and (4a). The actual situation is that parents expect to receive a fixed return from their children in the future without extravagant expectations. Transfers are only a fixed increase in the budget constraint. We illustrate the future return value of parents’ decisions to invest in education. The expectations of children can stimulate their current consumption choices. Selfish family decisions can turn out to be inefficient.

## 4. Complex Situational Decision-Making Process

In real life, parents in different social environments and at different times change decisions to consume and invest in their children’s education. Three typical scenarios are intercepted from the process in Figure 2 for discussion. We assume the existence of transfers, which are proportional to children’s human capital.

In Figure 4, at stage $T_{1}$, parents have a stable income and children are in the pre-school stage. At stage $T_{2}$, children receive public schooling until graduation. At stage $T_{3}$, parents do not continue to invest in human capital when their children graduate and enter the workforce.

Children are at an early age, when the level of human capital is low and likely to remain unrevealed. Parents judge their children’s potential based on their own knowledge, self-awareness, and social experience and thus choose the investment path. At this point, from a utility perspective, parents tend to choose an investment ratio that satisfies Equation (17). Usually, the growth of children in this stage is more likely to satisfy their parents, and parents invest more in their children and consume less, reflecting the parents’ general care for their children.

Further, we split the schooling timeline to obtain scenario 2 in Figure 5. We discuss the lower stage ${T_{1}}^{\prime }$ and the upper stage ${T_{2}}^{\prime }$ at school. The difference between Scenario 2 and Scenario 1 is mainly human capital in the initial decision-making period. Individual competence is clearly better in the higher grades than in the lower grades, and normally $A_{1}<A_{2}$. Parents will expect future education outcomes for their children, and investment decisions are made using Decision 2. The investment ratio satisfies Equation (20). Parents will prefer to invest more in higher grades of education, given the total amount of investment.

The third scenario is pursuing children’s studies. We split stage ${T_{2}}^{\prime }$ into college ${T_{1}}^{\prime }$ and postgraduate ${T_{2}}^{\prime \prime }$ in Figure 6. The level of the human capital of children is at a higher stage, and the option of further education does not necessarily lead to higher output capacity. Investment in education may be less valuable than consumption. From the previous discussion of Equation (20), it is likely that parents in this scenario will not invest more in their children’s educational progress. Parents meet their children’s basic living expenses. Therefore, at this stage of decision-making, parents may be more inclined to self-consume for satisfaction. In fact, Scenario 3 reflects the fact that when children have the quality of independence and autonomy, the investment in education provided by parents does not have a desirable upgrading effect, and the growth of children at this stage comes from self-accumulation and self-learning.

The three scenarios correspond to three typical decision-making processes. Notice that the children in Scenarios 2 and 3 already have the capacity to act and present themselves, and parents can examine their children’s character and abilities. In other words, parents also judge the extent to which their children care about them and the extent to which they are willing to transfer payments in the future. Scenario 1 Parents may choose to use their own level of transfers for prediction purposes, or they may choose to make decisions without considering the factor because of future uncertainty, as a “bad child” may not be able to give transfers to their parents.

We analyze the dynamic decision-making processes of families during the growth of children. As the children grow older, the family’s education investment may be high at the beginning and low at the end, and the family’s consumption may rise further. The reasons may be the lower marginal utility of the human capital level of older children and the higher cost of education. It can be seen that the investment decision of parents in the early years of their children’s development is in favor of education having a high future value. Despite the fact that higher education has a higher level of income, the return on its utility is not as high. Increased investment in education does not occur if parents do not have higher expectations. Thereby, there can be a complementary explanation for the higher weight of investment in the previous period. From the point of view of the children, the investment in education is acquired in order to accumulate more human capital. The findings of this study can explain the choice of households on the amount of total educational investment while maximizing children’s human capital. These two points have important value for discussion.

In summary, uncertainty and dynamic changes exist in children’s future potential, learning, personality characteristics, and other aspects. There is a dynamic change in the level of expected return on parental investment and the evaluation of children’s abilities. Thus, at different times, parents’ investment and consumption choices are based on a combination of factors such as the current social environment, experience, and knowledge. And parents’ decision-making is constantly changing and adjusting, it depends on the marginal utility of their consumption or investment.

## 5. Discussion

The results of parental decision-making are discussed on a composite timeline for three stages, including early childhood, schooling, and further study in higher education. The model conclusions provide a realistic analysis of the actual investment paths that parents might choose to take. These conclusions are also in line with the discussion in most of the literature [4,11,14,17].

Parental self-consumption and investment in children’s education are discussed and analyzed from a micro-family perspective. The conclusions obtained have important theoretical value for the study of intergenerational family consumption and investment behavior. At the same time, the dynamic mechanism of parents’ choice of consumption and investment in education is also a novel angle to study the growth and education of children. Based on the above conclusions, enlightening suggestions are made for research on education, development, and policy.

First, in order to incentivize families to invest in education to promote the growth of their children, it is necessary to analyze the path of investment decisions of families in the current period. The government can provide parents with the right guidance at the cultural and intellectual levels to set up a correct concept of education investment and choose reasonable investment methods. By reducing behaviors such as inefficient investment in education and overinvestment, children can receive adequate and appropriate education at all stages for healthy growth.

Second, an important factor in urban-rural inequality in education is the relative difference between income levels and the price level of education. For rural, low-income households, education subsidy policies can fall short of expectations because households choose to consume more. As the average level of human capital increases, the willingness of rural households to invest in their children’s education may decline because the burden of payment is too high [21,23]. The government needs to increase the ability of rural areas to pay for education and reduce the burden on households to pay for education, while at the same time inducing the public to accept the benefits of education. Then, households will not only be more willing to invest in their children’s education but will also have a higher budgetary capacity to spend on their children’s education.

Third, consumption and monetary policies have an impact on the investment decisions of households. The interplay between policies that increase personal budget constraints and enhance the ability of households to pay for consumption and education policies can incentivize an increase in the level of household investment in education.

## 6. Conclusions

This study discusses the decision-making behavior of parents between choosing self-consumption and investing in their children’s education. The value of the decision is to maximize utility and to achieve the optimal level of personal consumption utility as well as the optimal children’s human capital expectations, subject to basic constraints.

First, we constructed a continuous utility model, with Decision 1 focusing on the growth value of inputs in each period and Decision 2 focusing on the final value of children’s human capital. The two decisions correspond to different objectives, Equations (4) and (4a). Two important conclusions were obtained through the maximum value principle. Conclusion 1 is that the rate of growth of consumption is independent of the path of investment decisions, and Conclusion 2 is that, in general, the two decisions for a given level of investment will not result in identical investment paths.

In order to be used to interpret real life, we used a three-period logarithmic utility model for intertemporal discussion of decision choices and decision behavior. Parents will make a trade-off between utility losses and losses in their children’s future human capital. Higher levels of initial human capital will dampen parents’ desire to invest because the return on investment is lower and parents will choose to consume more for themselves. It also shows that early investment is more valuable, which is also consistent with the empirical results [11,21].

Considering transfer payments made by children in the future deepens the theoretical significance of the model. The presence of transfer payments leads to better options for parents, not only as an altruistic factor that improves parental care for their children but also as an increase in the level of the total budget. Parents can obtain more future transfer payments by investing in education, thus increasing the level of self-consumption. These analyses support the conclusions of the literature [22,24]. Conclusion 3 specifically states that when parents perceive the existence of future transfers proportional to their children’s human capital, parents increase upfront consumption and investment in education to increase the level of total utility.

For further discussion, debt financial instruments should be considered. The consumption constraining conditions of wages in each period are more stringent. The crowding-out effect of consumption on education investment will be more obvious when parents invest not only capital but also their leisure time. The utility analysis can explore how personal preferences affect the way children grow up, expanding the structure of the accumulation function.

## Figures and Tables

**Figure 1 behavsci-14-00224-f001:**
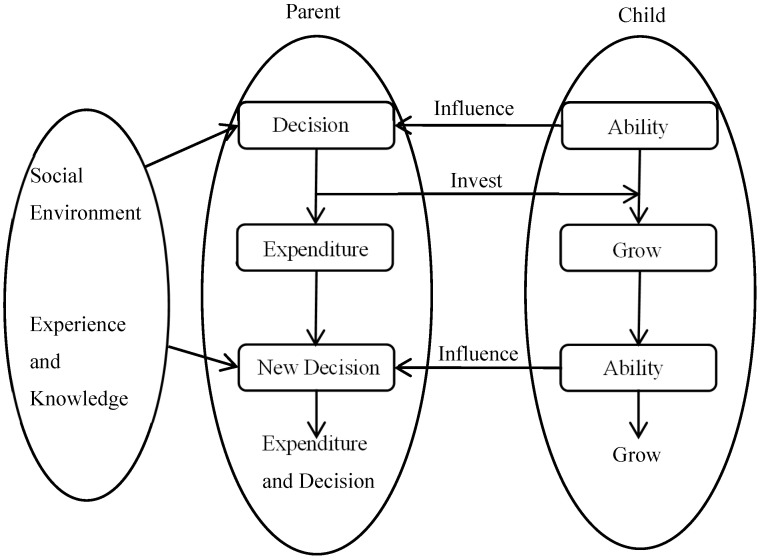
The cycle of consumption and investment in a household decision-making process.

**Figure 2 behavsci-14-00224-f002:**
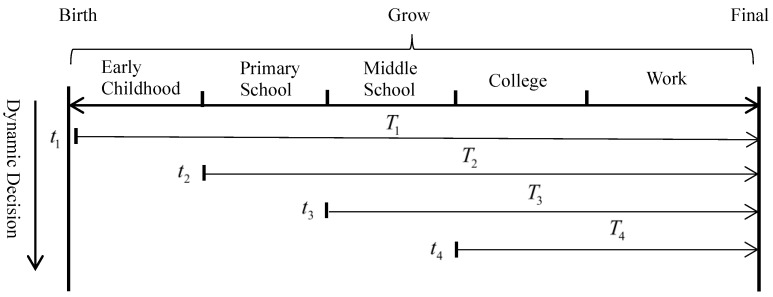
Dynamic decision-making processes at different times in a child’s life.

**Figure 3 behavsci-14-00224-f003:**
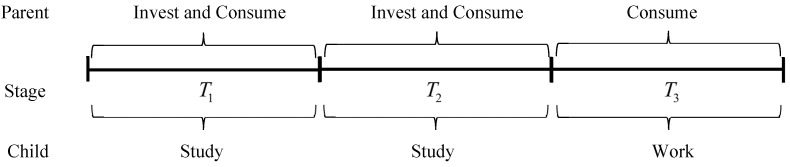
A three-stage growth process, including the child’s development and parental behavior.

**Figure 4 behavsci-14-00224-f004:**
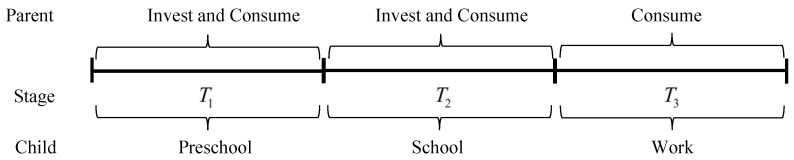
Scenario 1 describes a child growing up from infancy to adulthood.

**Figure 5 behavsci-14-00224-f005:**
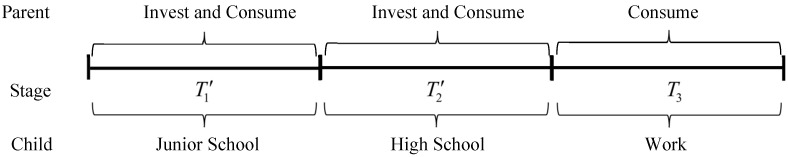
Scenario 2 describes the process of children’s access to schooling.

**Figure 6 behavsci-14-00224-f006:**
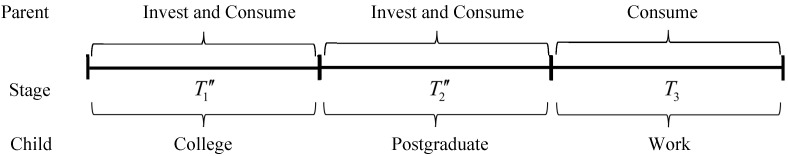
Scenario 3 describes the process of furthering children’s education.

## Data Availability

No new data were created or analyzed in this study. Data sharing is not applicable to this article.

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
