# Peer review of "Household Decision-Making Choices: Investment in Children’s Education or Self-Consumption"

_behavsci, 2024, doi:10.3390/bs14030224_

Round 1

Reviewer 1 Report

Comments and Suggestions for Authors

1.  The paper develops a theoretical and conceptual proposition for modelling the parental decision process on the choices between self-consumption and investment in children’s education. This theoretical/conceptual character of the paper should be strongly marked in the title of the article, in the abstract and introduction.

 2.       The proposed model solutions/approaches/equations should be more closely related to economic theories. Some statements describing the model appear to be arbitrary (part 3 of the paper). A greater connection between the model assumptions and economic theory (with more frequent references to economic theory) would enhance the scientific quality of the study.

 3.       The limitations of the proposed model should be indicated and discussed. For example, the models omit the possibility of using debt financial instruments or accumulated assets, which modifies the trade-off choices between consumption and expenditure on children's education.

 4.       The language of the study in some parts of the article is not clear and precise enough. In this respect, the quality of the language requires verification. For example:

-          Lines 10/11: “ This study uses the continuous time utility model to demonstrate the independent characteristics of consumption structure and education investment structure”.. (the discussion refers rather to the level of consumption and the level of investment in education (the structure of family expenditure in this respect), than to the structure of the consumption or the structure of family’s investments)

-          Lines 28/29: “Individuals acquire knowledge and abilities through education to form human capital exceeds the general labor capacity and investment time to invest in economic activities”

-          Line 34/35:  “… in terms of the height of their children's future growth”

-          Lines 559-560: the same sentence is repeated

-          Line 573: “The chapter discusess..” (is the paper a part of a book?)

 -          The main objective of the study should be corrected. The objectives of the study are to analyse… Analysis is one of the research methods, one of the ways of achieving research goals.

5.       The explanations of the symbols used in the presented equations should be verified. It happens that not all symbols that appear in equations are explained directly below the equation. For example, in Eq. 3 – r is not explained.

Comments on the Quality of English Language

1.       The language of the study in some parts of the article is not clear and precise enough. In this respect, the quality of the language requires verification. For example:

 -          Lines 10/11: “ This study uses the continuous time utility model to demonstrate the independent characteristics of consumption structure and education investment structure”.. (the discussion refers rather to the level of consumption and the level of investment in education (the structure of family expenditure in this respect), than to the structure of the consumption or the structure of family’s investments)

-          Lines 28/29: “Individuals acquire knowledge and abilities through education to form human capital exceeds the general labor capacity and investment time to invest in economic activities”

-          Line 34/35:  “… in terms of the height of their children's future growth”

-          Lines 559-560: the same sentence is repeated

-          Line 573: “The chapter discusess..” (is the paper a part of a book?)

Reviewer 2 Report

Comments and Suggestions for Authors

I find that the topic is interesting, and the paper is well written. The research model used by the authors is appropriate. While this paper is well written, I have one comment for the authors to make this paper more valuable. 

I believe that the relationship between how much parents value the importance of children’s education relative to parents’ entertainment and parents’ utility (well-being) must be an important link in the decision-making. So, given the current method used by the authors, how do the authors model the relationship between parents valuing the importance of children’s education relative to parents’ entertainment and parents’ well-being and investment behavior in education?

Reviewer 3 Report

Comments and Suggestions for Authors

I enjoyed reading your paper on "Household Decision-making Choices: Investment in Children’s Education or Self-consumption", analyzing how parents' investment in their children's education changes as the children grow older and mature.

The paper is well-written with only a few grammatical errors. I recommend accepting this article once the minor points are addressed.

In the “1. Introduction” it is crucial to include some more justifications highlighting the significance of comprehending household choices regarding self-consumption versus investing in children's education, with a focus on the impact on educational policies and economic progress.  It would be nice to provide a concise summary of the current literature on household economic behavior and explain why this research is significant for families, policymakers, and the broader community. By including these components in the introduction, you offer readers a clear outline of what to anticipate from the study, its relevance, and its contribution to the field's comprehension of household economic decision-making. 

To improve the section on "2. Characteristics of consumption and investment decisions in households," it would be useful to include a brief summary of the seminal literature that has informed current understanding of these dynamics.

The 3. Modelling and analysis” approach is robustly detailed, ensuring the credibility of the conclusions.

The 5. Discussion” could benefit from incorporating relevant literature to provide context for the findings and connect them to the wider field.

Good luck to your work!

Comments on the Quality of English Language

The paper is well-written with only a few grammatical errors.

Round 2

Reviewer 2 Report

Comments and Suggestions for Authors

The authors responded to my concerns, but their response makes me a little confused. 

The authors stated, “Because utility functions are typically separable, the entertainment utility component can also be separated from the consumption utility.” Why? Would you explain more clearly? Then the authors stated, “However, this has no particular effect on the decision process of spending or investing in education.” Can any of your equations shown in the text demonstrate your statement? Would you show readers more clearly?

If the authors are not able to demonstrate what I request, the authors may use a different way to demonstrate my concerns to convince readers. The authors need to include my concerns in the text and use a convincible way to convince readers.
